# Integration of Ultrasound in Image-Guided Adaptive Brachytherapy in Cancer of the Uterine Cervix

**DOI:** 10.3390/bioengineering11050506

**Published:** 2024-05-17

**Authors:** Elena Manea, Elena Chitoran, Vlad Rotaru, Sinziana Ionescu, Dan Luca, Ciprian Cirimbei, Mihnea Alecu, Cristina Capsa, Bogdan Gafton, Iulian Prutianu, Dragos Serban, Laurentiu Simion

**Affiliations:** 1Department of Radiotherapy, Regional Institute of Oncology, 700483 Iasi, Romania; drelenamanea@proton.me (E.M.);; 2“Gr. T. Popa” University of Medicine and Pharmacy, 700115 Iasi, Romania; 3“Carol Davila” University of Medicine and Pharmacy, 050474 Bucharest, Romania; ionescu_sinzy@yahoo.com (S.I.);; 4General Surgery and Surgical Oncology Department I, Bucharest Institute of Oncology “Prof. Dr. Al. Trestioreanu”, 022328 Bucharest, Romania; 5Radiology and Medical Imaging Department, Bucharest Institute of Oncology “Prof. Dr. Al. Trestioreanu”, 022328 Bucharest, Romania; 6Department of Morpho-Functional Sciences I—Histology, University of Medicine and Pharmacy “Gr. T. Popa”, 700483 Iasi, Romania; 7Surgery Department IV, Bucharest Clinical Emergency Hospital, 050098 Bucharest, Romania

**Keywords:** cervical cancer, cancer, brachytherapy, ultrasound, oncology, surgical oncology, image guided adaptive brachytherapy, radiotherapy, gynaecology, cancer treatment

## Abstract

Cervical cancer continues to be a public health concern, as it remains the second most common cancer despite screening programs. It is the third most common cause of cancer-related death for women, and the majority of cases happen in developing nations. The standard treatment for locally advanced cervical cancer involves the use of external beam radiation therapy, along with concurrent chemotherapy, followed by an image-guided adaptive brachytherapy (IGABT) boost. The five-year relative survival rate for European women diagnosed with cervical cancer was 62% between 2000 and 2007. Updated cervical cancer treatment guidelines based on IGABT have been developed by the Gynecological working group, which is composed of the Group Européen de Curiethérapie–European Society for Therapeutic Radiology and Oncology. The therapeutic strategy makes use of three-dimensional imaging, which can be tailored to the target volume and at-risk organs through the use of computed tomography or magnetic resonance imaging. Under anaesthesia, the brachytherapy implantation is carried out. Ultrasonography is utilised to assess the depth of the uterine cavity and to facilitate the dilation of the uterine canal during the application insertion. In this study, we examine data from the international literature regarding the application of ultrasound in cervical cancer brachytherapy.

## 1. Introduction

According to World Health Organization (WHO)’s reports [1], in 2022, cervical cancer accounted for around 660,000 new cases and approximately 350,000 fatalities (see Figure 1).

Cervical cancer has the most outstanding rates of occurrence and death in nations with low and intermediate incomes. This highlights significant disparities caused by the limited availability of national the human papillomavirus (HPV) vaccine, cervical screening, and treatment programmes, as well as social and economic factors that influence health outcomes.

Cervical cancer is a result of a long-lasting infection with HPV, and the implementation of prophylactic HPV vaccination and conducting screening and treatment for pre-cancerous lesions are highly successful [2] and economically efficient methods for preventing cervical cancer [3,4]. Vaccination not only provides protection to individuals who receive the vaccine against specific HPV types, but it can also decrease the occurrence of these targeted types in the general population. This reduction in prevalence leads to a decrease in infection rates among individuals who have not been vaccinated, a phenomenon known as herd protection or herd immunity. According to a recent model [5], it was projected that the objective of eradicating cancer might potentially be accomplished in low- and middle-income countries (LMICs) by attaining a 90% adoption rate for vaccination and implementing a screening programme that occurs twice in a person’s lifetime by the end of the 21st century.

Early detection and rapid treatment may lead to a successful cure for cervical cancer. Gynaecological malignancies were initially treated with brachytherapy in 1960 [6]. Afterwards, cervical cancer treatment has always included brachytherapy. Brachytherapy after chemoradiation is a conventional cervical cancer treatment. The reasons why these treatment methods are so important and have become a standard are the following. External beam radiation therapy, which is administered externally, is used to treat a broader specific region: cervical cancer itself and possible sites of cancer spread to other tissues, such as the lymph nodes. Brachytherapy is a therapeutic procedure that involves the targeted delivery of a minuscule radiation source directly into the cancer. This enables the administration of a concentrated amount of radiation to the cancer while minimising the exposure of nearby normal organs, which are exposed to a significantly lower amount of radiation. Another feature of brachytherapy is that it enhances the likelihood of achieving a cure while diminishing the potential harm to adjacent healthy tissues in proximity to the cancer.

Concurrent chemoradiation [7] is the ideal treatment for some stages of cervical cancer when radiation and chemotherapy are administered simultaneously. Chemotherapy enhances the efficacy of radiation therapy. Concurrent chemoradiation options include administering cisplatin on a weekly basis with radiation treatment. This medication is administered intravenously (IV) before the radiation session. If cisplatin is deemed unsuitable, carboplatin might be used as an alternative.

The administration of cisplatin in combination with 5-fluorouracil (5-FU) occurs at three-week intervals along with radiation therapy. The development of dose–volume histogram analyses in the previous decade has boosted the dosage to the target volume and minimised the doses to the organs at risk (OARs) [8]. The approach has improved from two dimensions (2D) to three dimensions (3D) using image-guided adaptive brachytherapy (IGABT), which is the standard cervical cancer treatment for many radiation oncologists worldwide, regardless of whether magnetic resonance imaging (MRI) or computed tomography (CT) scans are used (see Figure 2 showing the proportion between brachytherapy units and radiotherapy units worldwide). The GYN GEC-ESTRO working group (The Groupe Européen de Curiethérapie (GEC) ESTRO gynaecology (gyn) working group) (Appendix A) created target volume ideas in three dimensions, and the International Commission on Radiation Units and Measurements (ICRU) published the recommendations and encouraged IGABT implementation in many institutions. This approach has several advantages, including precise positioning of the applicator, enhanced treatment planning optimisation while maintaining target coverage, and reduced radiation dosage to organs at risk (OARs) [9,10,11,12,13,14,15,16,17].

The use of IGABT outcomes data enables the augmentation of target volume dosage while reducing the dose to normal tissue. Administering doses above 80 Gy, with a physiologically equivalent dosage of 2 Gy per fraction, for the D90 of the high-risk clinical target volume (CTVHR) results in a local control rate of around 90%. The retrospective EMBRACE research, which included 592 patients, reported a local control rate of 95% when the D90 to the CTVHR exceeded 92 Gy [9,10,11,12,13,14,15,16].

The use and advancement of image-guided adaptive brachytherapy methods in cervical cancer patients prompted a reassessment of the use of radiation inserts, as it can be seen in Figure 3 and Figure 4. Concurrently, a relationship has been demonstrated between dose–volume parameters and tumour response or the development of late radiation morbidity. Transabdominal ultrasonography (TAUS) is used to ensure the safety of the applicator insertion. Ultrasound (US) is a technique used to guide the dilatation of the uterine canal, measure the depth of the uterine cavity, and securely position the applicator.

TAUS is user-friendly, cost-effective, and can be accomplished with little exertion. TAUS provides therapeutic accuracy, consistency, and flexibility for each individual instance. Some brachytherapy planning methods include ultrasound as the only imaging modality. The abbreviation “US” is used during the insertion of the applicator and in the process of arranging therapy for cervical cancer, prostate cancer, and anal canal cancer. Additionally, it is beneficial for the insertion of brachytherapy catheters in cases of skin cancer, breast cancer, and head and neck malignancies.

Imaging advancements like 3D-US and contrast US surpass regular US in terms of providing more precise guidance for brachytherapy, resulting in enhanced accuracy. Van Elburg [19] states that 3DTVUS (three-dimensional (3D) transvaginal (TV) ultrasound (US)) imaging improves the visualisation of tumours and vaginal walls during gynaecologic perineal template interstitial needle insertion. Feedback from main users and a limited number of first patients has shown that this imaging technology is helpful for evaluating implants in a surgical environment.

In light of the current approach to brachytherapy for different types of cancer in the United States, and the urgent situation regarding cervical cancer in Romania, which ranks second in Europe in terms of both mortality and incidence rate, we conducted a comprehensive review of international databases to identify pertinent resources and assess the role of ultrasound in brachytherapy treatment for cervical cancer [20].

Romania is lagging behind several other European nations in its endeavour to eradicate HPV-induced malignancies. Approximately 3380 newly identified cases of cervical cancer are detected each year in Romania, as estimated for 2020. This disease ranks as the second most prevalent form of cancer among women aged 15 to 44 years. However, Romania has a very low rate of HPV vaccine adoption and cervical cancer screening. Only around 30% of eligible women get screened, and only 13% of teenagers receive immunisation. Another problem that puts Romania’s population in a very difficult place is a misconception about the cost of diagnostic and therapeutic procedures. Despite the fact that healthcare is free in Romania and the government provides new drugs, therapies, and screening measures through national programmes at no cost, the national screening and HPV vaccination programmes are not functioning properly. As a result, very few women in Romania undergo screening or receive HPV vaccination [21,22]. This scenario is well recognised and is accountable for the identification of the majority of patients in late stages, when radiation assumes a prominent position. WHO continues to emphasise the significance of cervical cancer and other illnesses caused by HPV on a global scale [18]. As a result, they strongly encourage the inclusion of HPV vaccines in national immunisation programmes, as well as the implementation of effective screening programmes [23]. As concerns the poor healthcare infrastructure, according to a report from 2021, Romania had 33 radiotherapy centres, the lowest number of CT scanners per 100,000 population (0.9) among the EU Member States, and just 0.4 MRI machines per 100,000 inhabitants [24].

## 2. Materials and Methods

After a previous initial search, we have updated our data by performing a more recent literature search (on 2 May 2024) on five different databases, as follows. (a) Search one was performed on www.Pubmed.gov (accessed on 2 May 2024) and retrieved 32 results after using the following as filters: English language, research conducted on humans, patients’ age between 19 years and >80. The 32 results consisted of 24 clinical trials and 8 reviews. (b) Search two was performed on www.scopus.com and returned 15 results, out of which 4 were reviews and 11 were articles, and the search on this site was limited to English language and to publications that were Journals. (c) Search three was conducted on www.sciencedirect.com and returned five results (one review article and four research articles). (d) Search four was conducted on www.academic.oup.com and retrieved only one result. (e) A fifth literature search was performed on www.reaxys.com and did not have any findings. On all five databases, the search terms were the same (as can be seen in Figure 5), consisting of the following: (((image guided brachytherapy or image guided adaptive brachytherapy or ultrasound guided brachytherapy or transabdominal ultrasound guided brachytherapy or IGABT)) AND (cervical or cervix)) AND (tumour or malignancy or cancer). The final number of articles included in the bibliography was determined by eliminating duplicates, unavailable material, and articles that did not address the exact subject.

When the articles found contained a subject that the authors deemed highly significant and pertinent to the current article, additional information on that specific matter was researched on Google Scholar, resulting in the inclusion of supplementary citations in the reference index.

## 3. Results

Brachytherapy is a very precise method of radiotherapy that achieves a conformal dose distribution by placing the radiation source in direct contact with the target volume using an applicator. Brachytherapy has been a significant treatment option for deep-seated tumours in soft tissue, the head and neck area, and breast for a considerable period of time, predating the introduction of external radiation methods and surgical procedures [25]. Despite a gradual and consistent decline in its significance, brachytherapy nonetheless remains crucial in the treatment of cervical malignancies. Considering the present state of cervical cancer in Romania, there is a revived interest in brachytherapy and it is a readily accessible and cost-effective alternative.

The radiation technique, first outlined in the early 1900s, relied on approximating the target volume by using a standardised form and pre-determined tabular data for the placement of applicators, source intensity, and duration of exposure. While the target volume received a sufficient radiation dosage, the organs at risk (OARs) were exposed to greater doses than required. The approach used two-dimensional X-ray projection pictures, which had limited planning capabilities and allowed for only limited modification of the source location [25]. Over time, various predictive dosimetry systems were created for both interstitial and intracavitary brachytherapy, including the Manchester [26], Manchester–Paterson–Parker [27], Quimby [28], Stockholm [29], Memorial, Fletcher [30], and Paris [31] systems. However, these systems were primarily based on clinical experience and basic dose calculation methods. These methods failed to consider the tumour’s response to radiation and the expansion and definition of tumour volume. As a result, they either administered insufficient radiation to a portion of the target volume or placed a greater burden on the surrounding organs at risk [32,33]. Despite their limits, these advancements in brachytherapy treatments represent significant progress towards standardisation and provide a means of verifying the effectiveness of new developments in radiation therapy [34,35]. Historical dosimetry systems have validated several technologies that greatly improved the treatment of tumours that require high doses of radiation, such as cervical or prostate cancers. These technologies include delivery systems like after loads, applicator technology, 3D visualisation using modern imaging with 3D target volume determination, and computerised algorithms for dose calculation [25].

In the past ten years, there have been improvements in the field of brachytherapy [36]. These advancements have allowed for higher doses to be delivered to the target area while minimising the doses to nearby organs at risk (OARs). This has been made possible through the use of dose–volume histogram analyses and the integration of different imaging techniques such as MRI, CT, US, and even positron emission tomography (PET). As a result, there was an improvement in comprehending the spatial distribution of the target volume, as well as tailoring treatment planning and documentation to each individual case [37,38,39,40,41,42,43].

Brachytherapy is crucial for treating locally advanced cervical cancer. In low–middle income countries (LMICs), the incorporation of imaging modalities for guiding brachytherapy is heavily reliant on the available infrastructure and resources. However, these resources may not always include access to MRI or CT scans, which are essential for this procedure [44]. Moreover, Eustace [45] found that trans-rectal ultrasound (TRUS) with CT guidance improves target delineation in underdeveloped nations without an available MRI.

Within LMICs, X-rays continue to be the primary method for guiding the insertion of applicators and assuring the accurate and secure positioning of the applicator in the uterine canal. While it is not acknowledged for its precision and features like MRI, CT, or other advanced imaging techniques (as shown in Table 1, which compares imaging modalities used for treatment planning in cervical cancer), it does not currently represent a first choice in recommendations for use in conformal planning [46,47]. Federico [48] observed that TRUS and MR were equivalent in determining preBT tumour maximum width in FIGO stage I/II cervical cancer. On the contrary, TRUS was marginally inferior to MR in advanced stages.

In LMICs, X-rays continue to be the most often utilised imaging technique for guiding the insertion of applicators and assuring the accurate and safe positioning of the applicator in the uterine canal. While ultrasound guidance is not as accurate or sophisticated as MRI or CT scans (see Table 2), it is more widely available and can be beneficial in treatment planning for cervical cancer, particularly when using advanced techniques such as three-dimensional contrast-enhanced ultrasound. However, it is important to note that specialised training is necessary for its use. Although it is not currently recommended in treatment guidelines for conformal planning, ultrasound guidance has the potential to improve outcomes [49,50,51]. Tharavichitkul [52] reported that CT-based and TAUS-based treatments had similar outcomes, but, nevertheless, the study’s retrospective analysis found greater vaginal toxicity with TAUS-BT.

The current imaging combination tactics for therapy monitoring comprise predominantly MRI/CT, US/CT, MRI/US, and MRI/PET. Two imaging technologies may be used to guide the implantation of applicators, rebuild applicators, shape the target and organs at risk (OARs), optimise dose, evaluate prognosis, and perform other tasks [53]. According to research conducted by Zhang, this technique provides a more suitable choice of imaging for brachytherapy [54]. Uterine cervix cancer patients may use (18)FAZA PET imaging. However, its predictive and prognostic utility is unclear, as shown by Schuetz [55]. The added utility of (18)FAZA-PET above morphologic repeated MRI in image-guided high-dose radiotherapy may help overcome hypoxia-related radio resistance.

## 4. Discussion

Brachytherapy practice in the US uses image guidance as a modality, which has the distinct benefits of being portable, cost-effective, and user-friendly. The United States authorised the use of precise, conformal, repeatable, and adaptive therapies.

The United States provides information on the varying echogenicity of tissues, aiding in the delineation of tumours and differentiation from the adjacent normal tissues. In the new planning system, brachytherapy only utilises ultrasound as the imaging modality.

The European Society for Radiotherapy and Oncology (ESTRO) recognises the significance of ultrasound (US) as an imaging technique in the brachytherapy process at three specific stages [56]: image-assisted temporary treatment planning, image-guided application, and image-assisted final treatment planning. However, the use of imaging in the brachytherapy procedure involves four distinct phases. In addition to those acknowledged by ESTRO, image-assisted quality control is included.

Currently, the United States employs brachytherapy in several forms of cancer, with particular focus on its potential use in gynaecological cancer [57]. Gynaecological brachytherapy has been used in the United States since 1970. US was effectively included into treatment planning for intracavitary applications [58,59].

TAUS is used in real-time during brachytherapy insertion to position uterine tandem and ovoids accurately and enable the assessment of thickness. This approach ensures a high level of probability and safety when using tandem and ovoids. Transabdominal ultrasonography is used to ascertain the dimensions, configuration, width, and diameter of the uterus, cervix, and cervical cancer, as well as the extent of parametrial involvement, as shown by Rao [60]. The use of intraoperative ultrasound guidance for tandem implantation reduces the likelihood of uterine perforation and enhances the ability to locally manage the tumour. The advantage was that ultrasonography allowed for direct visualisation of the uterus and provided real-time input to assist in choosing an appropriate tandem length and curvature. Uterine perforation is a very serious and unwanted complication which might result in bleeding and, also, in an extended duration of treatment [61] and reduced local control of the cervical malignancy, with improper dosimetry being sometimes unavoidable. Furthermore, a meta-analysis conducted by Sapienza [62] revealed that patients receiving ultrasonography-guided brachytherapy insertion had a 90% lower rate of uterine perforation per insertion than those receiving unguided insertion. Another important aspect that one can refer to in order to decrease the rate of uterine perforation is that the choice to use ultrasonography as a guide (in ovoid insertion) is tailored to each patient and based on the level of resistance encountered during cervical dilation, with insertion applied only by highly trained personnel (for instance, gynaecologic oncologists), as suggested by Bayrak [63]. Moreover, another method of protection against uterine perforation is the use of a Smit Sleeve (defined as a disposable intrauterine tube that is inserted into the cervical internal os to facilitate precise surgical localisation in cases of cervical cancer). The item is made from polymers and equipped with wing extensions that aid in its stable positioning for the purpose of identifying landmarks for future treatment. Ultrasound also gives information on how to position the applicator in different regions, as shown in Figure 6.

The technology (as described also by Salas [64]) enables the capture of a series of transabdominal axial pictures using ultrasound. These images may be submitted to a treatment planning system, where the high-risk clinical tumour volume (HR-CTV) and risk organs can be outlined. Subsequently, a dose delivery plan is constructed with orthogonal X-ray pictures taken at 0° and 270° angles, employing the International Commission on Radiation Units and Measurements (ICRU) 38 points. The same approach is used in both the series of ultrasound pictures and their corresponding volumes, and any discrepancies are then recorded. Therefore, the data obtained from the US may be verified using orthogonal radiographs to determine the dosimetry for the bladder and to calculate the real-time impact on the surrounding organs.

The process of brachytherapy insertion involves the placement of the applicator inside both the cervix and the uterine cavity. When the application is performed without direction, it might result in a poor application. This may further lead to the insufficient coverage of the cancer, suboptimal management of the malignancy at the local level, and increased radiation doses to the organs that are at risk. Additionally, it might lead to uterine perforation, which is an undesirable outcome. While not a widely used method, the instructions provided by the US for the administration of brachytherapy helps to minimise the hazards associated with incorrect placement. The radiation oncologist verifies that there is no perforation of the uterus. The sensation of resistance is perceived in the lower part of the uterus, indicating the presence of the applicator/tandem within the uterine cavity. The incidence of uterine perforation after blind insertion, as reported in research, ranged from 2% to 17% [65]. Pareek [66] performed a randomised controlled study of trans-abdominal ultrasonography for intracavitary brachytherapy (ICB) to minimise perforation and organ at risk dosages. Dosimetric testing indicated that US aid significantly reduced organ dosage and the conclusion was that for ICB treatments in resource-limited settings, trans-abdominal US should be standard.

Several investigations have identified additional risk factors for failed tandem insertion, including advanced age (over 60 years), previous surgical history, deformed anatomy of the cervix, retroverted uterus, and tumour development. Furthermore, after the administration of the first pelvic and tumour irradiation using external beam radiation treatment (EBRT), there is a possibility that the vagina and/or endocervical canal might experience full obstruction as a result of future tissue fibrosis. The prevalence of stenosis in women ranges from 59% to 88% [67], with complete stenosis seen in up to 11% of individuals. The task of inserting brachytherapy applicators into the uterine canal is challenging and carries a significant amount of danger.

Addley et al. [68] used a combination of real-time ultrasonography and direct hysteroscopy guidance to effectively implant the IUBT applicator in a patient diagnosed with stage IIa1 cervical SCC. This patient had a prior unsuccessful attempt at insertion due to significant stenosis of the endocervical canal. The conducted research demonstrated the successful navigation of post-radiation changes, hence averting complications and ensuring a positive outcome [46,69,70,71,72,73].

The first accounts of the use of ultrasound (US) in gynaecological treatment were in the context of interstitial brachytherapy. The research study had a sample size of 12 patients, and the authors established a correlation between the usefulness of ultrasound (US) in identifying the target and the organs at danger [74].

In addition, another study documented the use of transrectal ultrasonography (TRUS) to accurately position the needles inside the tumour volume, while avoiding any damage to the rectum [75,76,77] or bladder [78]. Researchers from various teams have shown the safety and efficacy of using ultrasound to guide the location of needles during brachytherapy implantation.

As shown by Potter et al. [79], chemoradiotherapy and MRI-based image-guided adaptive brachytherapy (IGABT) provide successful and consistent long-term local control in all stages of locally advanced cervical cancer, with few serious complications per organ. Furthermore, Mahantshetty et al. documented the use of TAUS and established a correlation between ultrasound (US) and magnetic resonance imaging (MRI) for treatment planning [79].

A research study including 292 patients with cervical cancer found that ultrasound (US) was the only strategy employed to aid in guiding and planning the treatment for intracavitary cervical brachytherapy. The local control rate over a period of five years was 87.5%, and there were no notable instances of toxicity [59].

The GEC-ESTRO asserts that ultrasound might be used for the precise positioning of needles and the delineation of the high-risk clinical target volume (CTVHR). Nevertheless, ultrasound cannot be reliably used in treatment planning for complex interstitial brachytherapy. The Vienna team asserts that in the future, TRUS has the potential to be used as a technique for defining the CTVHR [59].

Trans-cervical endo sonography (TRACE) is a novel technique used for pre-planning, real-time guiding, and treatment planning in cervical cancer patients undergoing image-guided adaptive brachytherapy (IGABT). This notion has the potential to enhance the current imaging techniques by providing further information [80].

Brachytherapy is a kind of radiotherapy that administers a concentrated dosage of radiation [81]. It is often compared to newer radiotherapy procedures like intensity-modulated radiotherapy (IMRT) and stereotactic body radiotherapy. It may also serve as a substitute for external radiation therapy. IMRT provides the opportunity to deliver a simultaneous integrated boost (SIB) to achieve a shorter treatment duration and a more uniform distribution of radiation dosage, while minimising excessive doses beyond the targeted area [82].

To effectively compete with advanced radiotherapy methods that administer high levels of radiation directly to the tumour while minimising radiation exposure to nearby organs and reducing uncertainties caused by movement, brachytherapy must be performed in accordance with recommended guidelines, utilising modern imaging techniques for guidance and application [83].

Nevertheless, the brachytherapy technique carries various risks, including the potential for needle displacement between application and treatment, the challenge of accurately delineating the tumour volume due to the presence of visible CT-caused metallic needles, and the possibility of bleeding by perforating vessels in close proximity to the tumour [84,85,86]. Kirchheiner [87] showed that several deficits in health-related quality of life and patient-reported symptoms were detected following definitive radio(chemo)therapy with IGABT, with varying patterns of improvement and indications of recovery three months later. Some functional health-related quality of life aspects remain affected. To prevent the occurrence of artefacts generated by metal needles on CT scans, it is advisable to use CT- and MRI-compatible applicators and needles made of non-metallic materials.

In individuals with minimal body fat, the intestines descend into the pelvic region, making it challenging to capture clear images owing to artefacts. Additionally, it becomes more arduous to distinguish the cervix and uterus. In this scenario, the bladder is presumed to be full of saline to eliminate the possibility of bowel injury in the event of uterine rupture.

During the transabdominal ultrasound (TAUS) procedure, the bladder is filled with physiological saline to optimise picture quality and place the uterus vertically. The volume of saline put into the bladder might range from 200 mL to 500 mL [88]. In individuals with an anteverted uterus, the bladder will be infused with a greater volume of saline, namely 400–500 mL, in order to reposition the uterus into a vertical orientation. The bladder fullness is determined based on the detected location of the uterus during the ultrasound examination. The pictures acquired by ultrasound are in the axial and sagittal orientations.

Obese patients also have difficulties, and the visibility of pictures provided by the US is diminished [89]. When dealing with a patient who has previously had a subtotal hysterectomy, it is crucial to acquire the highest quality ultrasound imaging possible due to an increased risk of perforation.

During interstitial brachytherapy, it is crucial to use ultrasound guidance while putting the needles at the level of the parametrium and paravaginal area. This helps prevent the needles from accidentally entering the bladder or colon. Using ultrasonography to guide the placement of needles for interstitial brachytherapy may provide valuable information, particularly when the ureters are dilated and can be seen.

Multiple researchers have examined the efficacy of using ultrasound (US) for the insertion of applicators and brachytherapy treatment for cervical cancer [90,91]. However, there is currently no established methodology for this specific purpose.

The Vienna working group assessed the efficacy of TRUS in the delineation of cervical cancer. The pictures acquired using transrectal ultrasound (TRUS) accurately locate the specific area of interest during brachytherapy. It was shown that there are restrictions in terms of the focus distance and vision field achieved with the endorectal probes [92].

Hu [93] performed a study to examine the effectiveness and safety of ultrasound CT-guided 3D intracavitary and interstitial brachytherapy (US-CT-3D-IGBT) for treating big cervical tumours with bleeding.

Pintakham [94] performed a study to evaluate the dosimetry comparison of brachytherapy delineation using two distinct methods: a volume-based plan utilising computed tomography (CT) and a point-based plan employing transabdominal ultrasound (TAUS) in CT datasets for brachytherapy. The results showed that the point-based plan performed by TAUS had better outcomes for both the intended targets and the organs at risk, in comparison to the volume-based plan carried out by CT. TAUS’s point-based technique led to a close closeness between D98 and HR-CTV.

Brachytherapy in prostate cancer uses ultrasound imaging to pre-plan treatment prior to application. This imaging is performed in real-time during needle insertion and is connected to CT planning. This approach has advanced the delineation in brachytherapy based on imaging. This notion was subsequently used in different types of malignancies. Brachytherapy in the US has used ultrasound (US) to assist with needle placement and delineation, often in conjunction with other imaging techniques.

Technological advancements in the field of medical imaging, such as the introduction of 3D ultrasound (US), contrast-enhanced US, and high-frequency US, have been incorporated into clinical practice to serve diagnostic, interventional, and therapeutic purposes. The 3D interventional treatments in the US have the advantage of accuracy and the potential for visualisation in many dimensions. The use of 3D ultrasound technology reduces the reliance on human operators and enhances the ability to replicate results [95].

Mastering and managing US imaging is not too challenging. In a research study conducted in India, 2-month surgery trainees who received training in US abdominal manoeuvres had a comparable understanding to radiology residents.

Although there are established guidelines for brachytherapy dose and fractionation protocols, there is a lack of data about the actual patterns of use. The aim of a recent study conducted by Hsieh [96] was to evaluate the current utilisation of cervical cancer brachytherapy in the United States and its association with patient demographics and facility characteristics.

The use of 3D ultrasound (US) in the context of cervical cancer for delineation is constrained by the challenging visualisation of organs at risk and the restricted ability to reproduce results consistently. Further research is needed to confirm the validity of these problems in cervical cancer as well as other forms of cancer. Scientific advancements will surpass these technological restrictions, considering the success achieved in prostate cancer.

Three-dimensional printing is a practical option in brachytherapy and teletherapy for electron and photon treatments because it improves the quality of treatment delivery by allowing for larger doses to be administered to target volumes. In a recent study conducted by Lee [97], it was shown that there is potential to reduce the need for repeat computed tomography (re-CT) and effectively improve time management for both patients and oncologists.

Another crucial aspect to consider in the discourse around brachytherapy is its heterogeneous use, which is impacted by several circumstances. Kumar’s study [98] revealed a significant decrease in the use of brachytherapy for cervical cancer. This drop has led to a decrease in both local control and overall survival rates. In addition, many national- and state-level databases have shown a substantial disparity in the accessibility to healthcare services and the use of BT boost among individuals diagnosed with cervical cancer in the United States. Referring to the simultaneous integrated boost (SIB), Lőcsei [99] showed that SIB is suitable for decisive RCTs in cervical cancer patients. If BT is not possible, SIB dose escalation may safely be given to cervical cancer patients throughout final RCT.

Regarding the future prospects of brachytherapy for cervical cancer, there are two ongoing clinical studies. The first study [100], named "Phase 0 Clinical Trial of Molecular Biomarkers in Women With Uterine Cervix Cancer," is a clinical trial in the first phase (phase 0) that aims to investigate molecular biomarkers in women who have been diagnosed with uterine cervical cancer. Women receive conventional radio-chemotherapy treatment, which is then followed with brachytherapy. Collecting blood samples is essential for detecting the presence of deoxyribonucleotides, human papillomavirus DNA, and circulating cancer cells. A study named “Comparison of Clinical Response and Toxicity of Hypofractionated Chemoradiation With Standard Treatment in Patients with Uterine Cervix Cancer” was conducted in the scientific environment described below: The treatment for uterine cervix cancer requires definitive therapy by the use of concurrent chemo radiation. This treatment combines external beam irradiation and chemotherapy, followed by high-dose-rate brachytherapy. Increasing the dose per fraction of treatment may reduce therapy time, hence minimising costs and patient exposure. The research seeks to determine if low fractionated radiation is not inferior to standard therapy.

Summarising the importance of ultrasound in needle placement, we cite the work signed by Knoth [101] which had a goal of examining the appearance of interstitial needles on transrectal ultrasonography (TRUS) in patients undergoing cervical cancer brachytherapy and analysing its influence on the quality of the implant and treatment plan.

Regarding the importance of ultrasound in treatment planning, we cite the work by Wang [102], who conducted a review of papers on various imaging techniques used for target delineation and planning. Advanced methods such as real-time image guiding and 3D printing were also included into IGABT. The author further presented a comprehensive review of the data on the practicality of these methods and focused specifically on the results seen in clinical settings. The conclusion was that, after demonstrating successful clinical results, the future of brachytherapy (BT) for locally advanced cervical cancer is anticipated to focus on improving the accuracy and efficacy of image-guided operations. Furthermore, it is crucial to reach an agreement on operational matters and promote technical progress in order to address the inherent limits of various imaging techniques.

An important matter regarding the research study design, and implicitly, its findings, is the fact that the elderly have therapeutic problems due to clinical trial underrepresentation of patients over 65. Venkatesulu [103] considered that image-guided brachytherapy may improve cervical cancer outcomes in older patients and Hata [104] concluded that even older people may benefit from curative radiation therapy.

At the end of this section of discussions regarding the treatment of cervical cancer and IGABT, we consider it essential to underline that recurrent vaginal forms of cervical cancer can be also treated successfully via a similar therapeutic intervention [105,106,107].

Another aspect that needs to be emphasised is the treatment’s impact on the lymph nodes in cervical cancer. Li [108] concluded that concurrent CT-guided 125I seed implantation and chemotherapy are more effective, safe, and painless than standard chemotherapy for cervical lymph node metastases.

The trend in treatment personalisation was also mentioned by Castelnau-Marchand [109], who presented preliminary studies that suggested adapting planning goals to specific clinical conditions and cofactors.

## 5. Conclusions

Medical imaging improves treatment results, ensures accurate and fast diagnosis, and directs treatment choices. However, two-thirds of the globe lacks diagnostic imaging. Recent research studies found a severe imaging equipment deficit in LMICs that have fewer than one CT scanner per million people, whereas HICs (high-income countries) have approximately 40. MRI and nuclear medical equipment are much rarer. Ultrasound can aid resource-constrained planning and this economic asset comes with many other advantages, because (a) it offers real-time feedback and is portable and cheap; and (b) it correctly portrays the uterus and cervix, enabling implant verification throughout the entire procedure. Furthermore, ultrasound-guided applicator insertions enhance implant quality. In this context, higher implant technical quality enhances local control and minimises toxicity.

Brachytherapy is a very precise method of radiation and this study emphasises the significance of implementing ultrasound (US) in brachytherapy for cervical cancer as a crucial component of an effective therapeutic approach. In the future, the IGABT technology will be expanded to more institutions, allowing for a much more significant inclusion of US in brachytherapy treatment for cervical cancer than what we see today. This expansion has the potential to enhance local disease management and minimise toxicity.

As an overview and brief conclusion of the difficult conditions in which Romania faces cervical cancer, we mention the following: a significant proportion of women delay undergoing regular medical examinations due to constraints in terms of time and financial resources. These studies also indicate that a lack of awareness about the disease and the specific preventive measures lead to low participation in screening and HPV vaccination programmes implemented in Romania, thereby exacerbating the country’s cervical cancer situation. We have seen that the national programmes exhibit intricate processes, suffer from inadequate funding, and fail to sufficiently incentivise healthcare staff. Additionally, the scarce data offered to the eligible population contribute to a very low rate of women being checked and vaccinated. The inadequate healthcare infrastructure, including insufficient treatment facilities and equipment, also leads to poor rates of treatment response and cures. Our conclusion is that the Romanian Ministry of Health should promptly take action by launching extensive awareness campaigns, establishing steps to ensure the functionality of the programmes and assuring consistent financing.

## Figures and Tables

**Figure 1 bioengineering-11-00506-f001:**
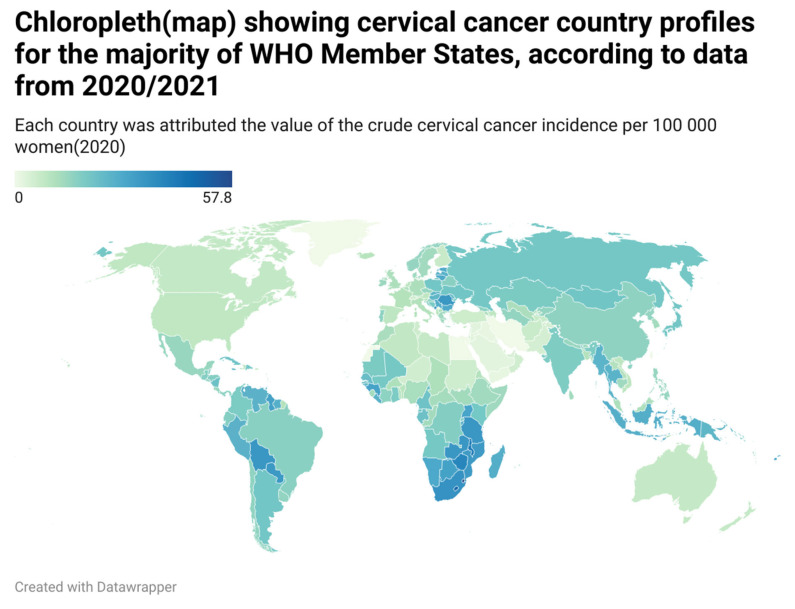
A Chloropleth map showing cervical cancer country profiles for the majority of WHO Member States, according to data from a 2020/2021 report.

**Figure 2 bioengineering-11-00506-f002:**
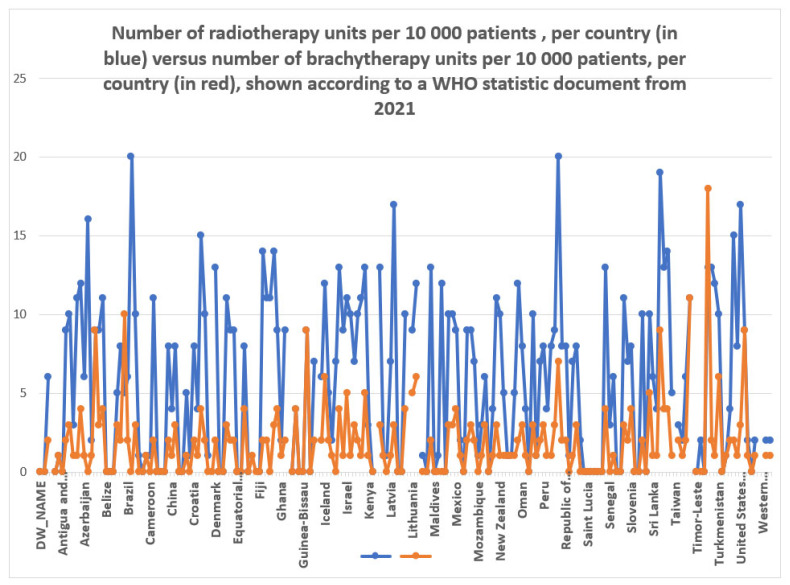
Comparison between radiotherapy units (in blue) and brachytherapy units (in red), shown per 10,000 patients, per country, according to a WHO report from 2021 [18].

**Figure 3 bioengineering-11-00506-f003:**
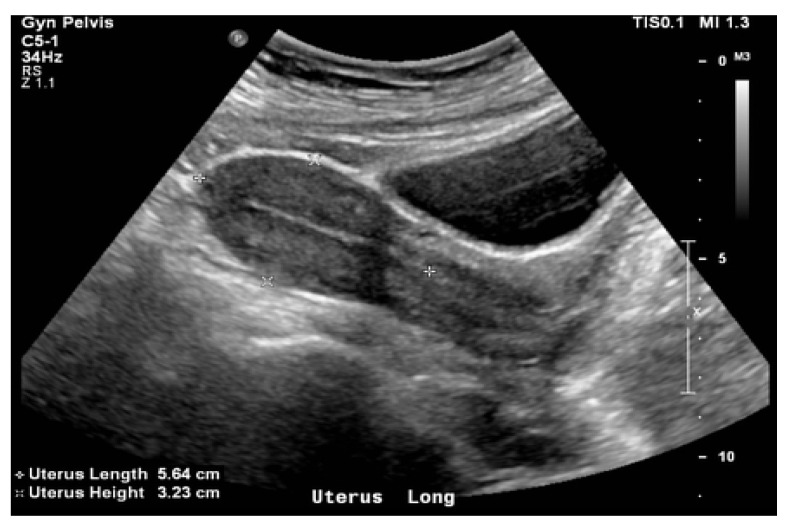
General aspect of the uterus and of the cervix during an intravaginal ultrasound.

**Figure 4 bioengineering-11-00506-f004:**
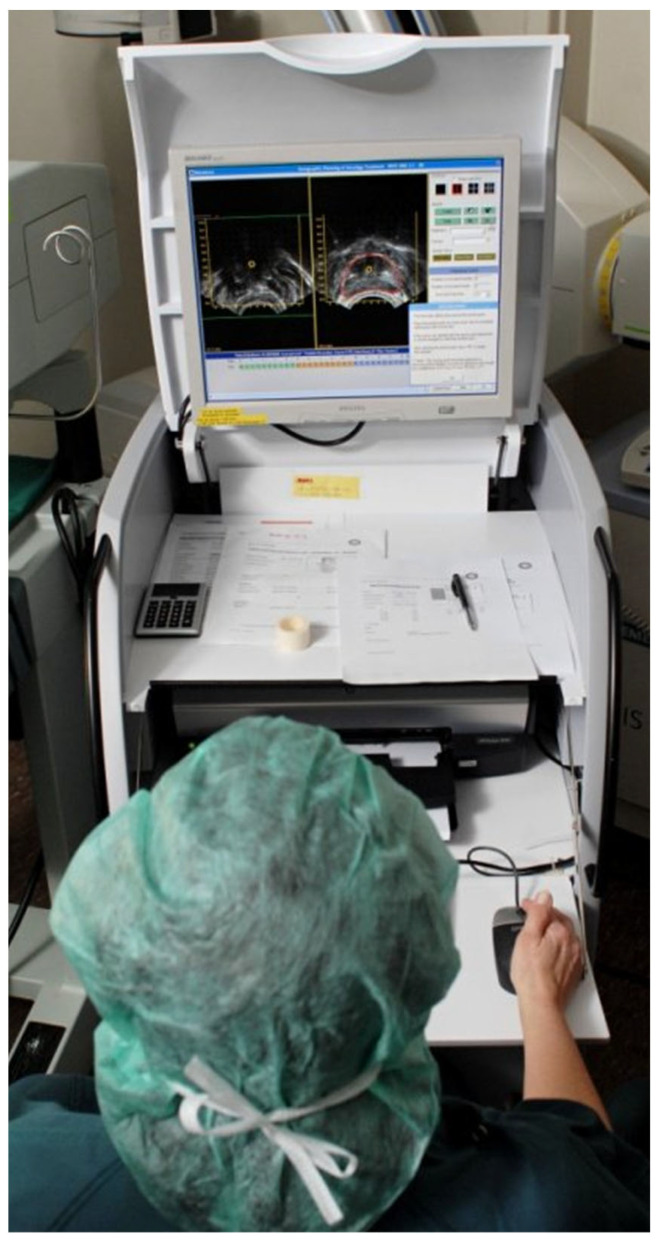
Treatment planning image of a patient treated with brachytherapy. The source of the picture is: Rock mc1, CC BY-SA 3.0 (https://creativecommons.org/licenses/by-sa/3.0) (accessed on 2 May 2024), via Wikimedia Commons.

**Figure 5 bioengineering-11-00506-f005:**
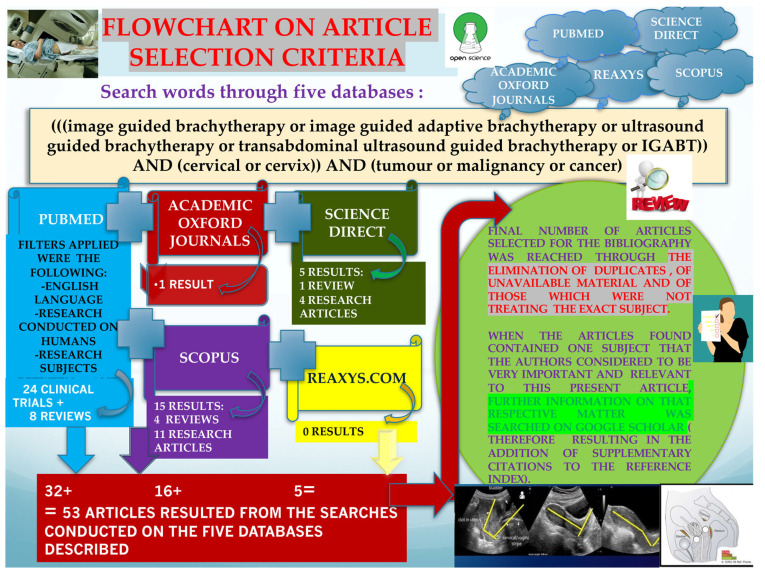
Flowchart depicting the literature research on the five databases mentioned and the criteria applied in order to obtain the final article selection.

**Figure 6 bioengineering-11-00506-f006:**
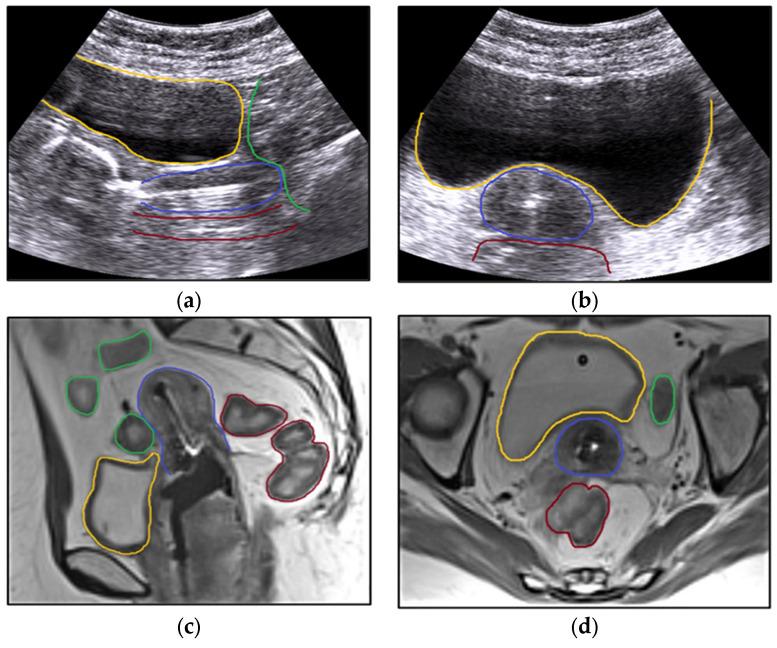
A clinical situation when the Fletcher applicator was used. Sagittal (**a**) and axial (**b**) view with the applicator in the uterine canal on transabdominal ultrasound and sagittal (**c**) and axial (**d**) pelvic MRI. The fundamental principles pertaining to the implantation of intracavitary brachytherapy are shown. Measuring the uterine canal and taking precautions to prevent perforation is advantageous, particularly when considering the organs that are vulnerable (**a**,**b**). MRI pictures provide a clearer identification of the relationship between the organs at risk (**c**,**d**). Organs at risk are the bladder (yellow), rectum (brown), and bowel (green). During brachytherapy insertion, the bladder is filled with 400 mL of saline to bring the uterus (blue) into a vertical position (**a**,**b**), while with MRI, the bladder is filled with 50 mL of saline (**c**,**d**). The applicator is fixed using vaginal gauze packing.

**Table 1 bioengineering-11-00506-t001:** A comparison of several imaging modalities used for treatment planning in cervical cancer.

Imaging Modality	X-ray	Ultrasound	Tomography (CT-SCAN)	Magnetic Resonance	Positron Emission Tomography
Soft-tissue resolution	Poor	Good	Good ^@^	Excellent	Good
Geometric accuracy	Good	Good	Good *	Good ^#^	Excellent
Image quality	Protocol dependent	Operator and protocol dependent	Sequence and protocol dependent	Sequence and protocol dependent ^&^	Sequence and protocol dependent ^&^
Artefacts	Metal	Multiple types and causes	Multiple types and causes	Multiple types and causes	Multiple types and causes
Slice orientation	Single planar	Multi-planar	Trans-axial	Multi-planar	Trans-axial
Usage in cervical cancer	Yes	Yes	Yes	Yes	Rarely used
Accurate visualisation and reconstruction of brachytherapy applicators	Yes	Yes	Yes	Yes	Yes
Types of applicators	Metal or plastic with metal X-ray guides	Metal, plastic	Metal, Plastic	Metal ^%^, Plastic	Metal, Plastic
Possibility of radiation dose calculation	No	NA	Yes	NA	Yes
Portability (potential for intraoperative use)	Yes	Yes	Sometimes available	Sometimes available	No
Time to obtain image	Seconds	Minutes	Seconds	Hour	Hours
Availability	Low	Low	Medium	High	High
Cost of equipment	Low	Low	High	High	High
Cost of scan	Low	Low	High	High	High

Table legend: ^@^ the use of contrast may improve the results; * it depends on the human aspect to accurately focus on the area of interest; ^#^ the precision diminishes as one moves farther from the magnet core; ^&^ needs sufficient patient compliance; ^%^ there are just a few kinds of metal that may be used owing to safety concerns with MRI; NA—currently, there are no accessible data. However, ongoing investigations are being conducted to provide estimations of electron density for the purpose of radiation applications.

**Table 2 bioengineering-11-00506-t002:** Advantages and limitations of CT and MRI in treatment planning.

CT versus MRI for Treatment Planning: Advantages and Limitations	
**CT advantages**	**CT limitations**
Since CT was the first 3D imaging modality, most treatment-planning algorithms were created for it.	Suboptimal tissue contrast.
More people utilise CT scanners than MR.	Lack of functional information.
CT has superior geometric fidelity than MR, which may distort the image.	Difficulty to see minuscule cancer cell groupings from the gross tumour.
Identifying the mass attenuation coefficient (μ/ρ (m^2^/Kg)) or attenuation characteristics for high-energy photons, X-rays, and gamma rays is crucial for accurate dosage estimation using CT.	
**MRI advantages**	**MRI limitations**
Better soft-tissue contrast than CT.	Image artefacts.
Enhances the ability to differentiate between tissue that requires treatment and tissue that does not.	Lack of tissue density information.
Nonionising radiation.	Relatively small field of view.

## Data Availability

Not applicable.

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
