# Peer review of "Integration of Ultrasound in Image-Guided Adaptive Brachytherapy in Cancer of the Uterine Cervix"

_bioengineering, 2024, doi:10.3390/bioengineering11050506_

Round 1

Reviewer 1 Report

Comments and Suggestions for Authors

This is a review paper on the use of ultrasound (US) in the brachytherapy treatment of cervical cancer. The manuscript needs extensive revisions before it will be suitable for publication.

Abstract:

The abstract should be intelligible on its own, without a reader having to read the entire paper. And in an abstract, we usually do not cite references—most of your abstract will describe what you have studied in your research and what you have found and what you argued in your paper, in this case, your review. The abstract should be self-contained and fully understandable without reference to other sources. In general, it is not necessary to use abbreviations in the abstract because the abstract is so short.

Page 1, Line 23. The word treatment is missing - external beam radiation (EBRT)

Introduction:

Page 2, line 43. Cervical cancer has the greatest rates of occurrence and death in nations with low and intermediate incomes. Reference?

Page 2, line 48. those who have HIV are six times more prone to developing cervical cancer in… Start with capital T for “Those”. Any reference for this statement?

Page 2, line 48/49. those who have HIV are six times more prone to developing cervical cancer in comparison to those who do not have HIV. Why HIV? The authors were explaining about HPV and suddenly HIV appears.

Page 2, line 50. Implementing prophylactic HPV vaccination and conducting screening and treatment for pre-cancerous lesions are highly successful and economically efficient methods for preventing cervical cancer. Reference?

Page 2, line 54. Gynaecological malignancies were initially treated with brachytherapy in 1960. Reference?

Page 2, line 59. Please write in full 2D, 3D, IGABT, MRI, and CT before using the abbreviations.

Page 2, line 62. Please write in full GYN GEC-ESTRO before using the abbreviations. Even though it’s written in the list of abbreviations in page 12, it should be introduced in the text.

Page 4, line 96. What is 3DTVUS? Even though it’s written in the list of abbreviations in page 12, it should be introduced in the text.

Page 5, line 115. Reference no [13] introduced after [24]. It was not mentioned at all before this. Why?

Materials and Methods:

Page 5, line 119. Between July 27, 2023, and December 14, 2023, we ran a survey of the worldwide database PubMed to find materials related to IGABT in cervical cancer. Why only PubMed database? How about Web of Science? Science Direct? Scopus?

Page 5, line 130. Additional searches were performed using the same criteria as before on the websites www.sciencedirect.com (yielded 4 results) and www.scopus.com (yielded 583 items - 334 articles, 70 books or book chapters, and 145 reviews). When? Within the same period?

Page 5, line 134. review, we: removed the duplicates and the articles which did not match the theme of…. Why there is colon after we?

Results:

Page 6, line 169. In the past ten years, there have been improvements in the field of brachytherapy. Reference?

Page 6, line 173. techniques such as magnetic resonance imaging (MRI), computer tomography (CT)…. The abbreviations were already written in page 2, line 59. Why now only write in full?

Page 6, line 183. Within low- and middle-income countries (LMIC), X-rays continue to be the primary…Redundant. LMIC was already introduced and written in full in page 6, line 178.

Page 6, Table 1. The spacing of the text in the table row e.g. andSequence and typesMultiple.

Page 7, line 198. In low- and middle-income countries (LMIC), X-rays continue to be the most often… Redundant. LMIC was already introduced and written in full in page 6, line 178.

Page 9, line 269. Addley [58] used a combination of real-time ultrasonography… should write as Addley et al because he was not the only author, there were M. Persic, R. Kirke, and S. Abdul as well.

Page 9, line 291. The GEG-ESTRO asserts that the United States might be used for the precise… What is GEG-ESTRO?

Page 9, line 300 - 313: Reference?

Page 10, line 329. Obese patients also have difficulties, and the visibility of pictures provided… Any reference to support this statement?

Page 10, line 338. Multiple researchers have examined the efficacy of using ultrasound (US) for the insertion of applicators and brachytherapy treatment for cervical cancer. Any reference to support this statement?

Page 10, line 361. Technological advancements in the field of medical imaging, such as the introduction of 3D ultrasound (US), contrast-enhanced US, and high-frequency US, have been incorporated into clinical practice to serve diagnostic, interventional, and therapeutic purposes. 3D interventional treatments in the US have the advantage of accuracy and the potential for visualisation in many dimensions. The use of 3D ultrasound technology reduces the reliance on human operators and enhances the ability to replicate results. Reference?

Page 11, line 372. The aim of a recent study conducted by Hsieh [75] was to evaluate… why ref [75]? What happened to ref [62]?

Page 11, line 383. In a recent study conducted by Lee [77]… What happened to references 64-76? Missing?

Page 11, line 388. a significant decrease in the use of brachytherapy (BT) for cervical cancer. Why only introduce the abbreviation now? At the end of the manuscript?

References:

The authors cited their own papers 8 times from 81 references, about 10% of total references.

A lot of redundancies – introduction of abbreviations but the authors wrote in full and re-introduce the abbreviations, again and again.

Comments on the Quality of English Language

Moderate editing of English language required

Author Response

Respected Reviewer1, please find the authors' answers to your suggestions in the attached file! Sincerely yours, the authors.

Reviewer 2 Report

Comments and Suggestions for Authors

The paper requires major revisions.

1. Can you provide more context on the significance of the study in the current landscape of cervical cancer treatment?

2. How did you select the data sources for your literature review, and what criteria were used to determine their relevance?

3. In the abstract, you mentioned the standard treatment for locally advanced cervical cancer involves external beam radiation and chemotherapy followed by brachytherapy. Can you elaborate on why this combination is considered the standard?

4. Could you explain the role of the GYN working group in formulating updated treatment guidelines for cervical cancer?

5. The abstract mentions the use of 3D imaging with MRI or CT in treatment planning. Can you discuss the advantages and limitations of each imaging modality in this context?

6. How does ultrasound (US) assist in the brachytherapy treatment process, particularly in comparison to other imaging modalities like MRI and CT?

7. What specific advancements in image-guided adaptive brachytherapy have prompted a reassessment of radiation inserts, as mentioned in the text? Add the reference paper to the site with [ PMID: 28533803, PMID: 34544468]

8. The paper discusses the situation of cervical cancer treatment in Romania. Can you elaborate on the challenges faced in Romania regarding awareness, prevention methods, and healthcare infrastructure?

9. Can you discuss the role of prophylactic HPV vaccination and screening in preventing cervical cancer, especially in low- and middle-income countries?

10. How do the imaging modalities used for brachytherapy planning differ in terms of accuracy, availability, and cost-effectiveness, particularly in LMICs?

11. In what ways does ultrasound guidance improve outcomes in brachytherapy treatment, especially in regions where advanced imaging modalities may be limited?

12. How does the use of ultrasound aid in assessing the thickness and positioning of the tandem and ovoids during brachytherapy insertion?

13. Could you provide more details on how ultrasound data are verified using orthogonal radiographs for dosimetry calculations and assessing impact on surrounding organs?

14. What measures are taken to minimize the risks associated with incorrect placement of applicators during brachytherapy insertion, particularly regarding uterine perforation?

15. Can you discuss the implications of uterine perforation after blind insertion and how ultrasound guidance helps mitigate this risk?

16. In the context of the study's findings, how do you propose addressing the challenges faced in regions like Romania regarding cervical cancer treatment and prevention?

17. What are the potential future directions for research or interventions based on the insights gained from this study?

18. How do you envision the findings of this study contributing to improvements in cervical cancer treatment practices globally, especially in resource-limited settings?

Author Response

Respected Reviewer2, Please find the authors' answers to your suggestions in the attached file. Sincerely yours, the authors.

Round 2

Reviewer 1 Report

Comments and Suggestions for Authors

The manuscript has been revised according to the suggestions and comments. However, there are a few revisions needed.

Abstract:

The abstract can be further improved.

Line 23. followed by image-guided adaptive brachytherapy boost (IGABT). The abbreviation should be placed after brachytherapy, not boost.

Line 25. Updated cervical cancer treatment guidelines based on IGAB have been developed by the Gynecological working group,… Why IGAB, not IGABT?

Introduction:

Page 1, line 38. Please write WHO in full before using the abbreviation, which was only introduced in page 6, line 148.

Page 2, line 45. This highlights significant disparities caused by the limited availability of national HPV vaccine, cervical screening, and… the abbreviation was only introduced in line 47.

Page 3, line 85. regardless of whether magnetic resonance imaging (MRI) or computed tomography (CT scan) are used…. The abbreviation should be CT only = regardless of whether magnetic resonance imaging (MRI) or computed tomography (CT) scans are used….

Materials and Methods:

Page 6, line 156. After an initial quest performed from July 27, 2023 to December 14, 2023, we have updated our data by performing a more recent literature research (on May 2, 2024) on five different databases, as follows: In my opinion, no need to mention the initial quest to 14/12/2023, but just mentioned the latest one until 2/5/2024.

Conclusions:

Page 15, line 542. Recent researches found a severe imaging equipment deficit in LMICs that have fewer than one computed tomography (CT) scanner per million people. The abbreviation was already introduced in page 3, line 86.

Please double check for all abbreviations used in this paper.

Comments on the Quality of English Language

The quality of English is improved.

Author Response

Respected Reviewer1, Please find our answers in the document attached! Thank you very much for your time and effort! Respectfully yours, the Authors.

Reviewer 2 Report

Comments and Suggestions for Authors

After a thorough evaluation of the revised manuscript, I am pleased to report that the authors have successfully incorporated all of the recommended changes, which have significantly improved the quality of the paper.

Author Response

Respected Reviewer2, Thank you very much for your time and effort! Respectfully yours, the Authors.